# The Relationship between Language and Technology: How Screen Time Affects Language Development in Early Life—A Systematic Review

**DOI:** 10.3390/brainsci14010027

**Published:** 2023-12-25

**Authors:** Valentina Massaroni, Valentina Delle Donne, Camillo Marra, Valentina Arcangeli, Daniela Pia Rosaria Chieffo

**Affiliations:** 1Department of Safety and Bioethics, Catholic University of Sacred Heart, 00168 Rome, Italy; valentina.massaroni@unicatt.it (V.M.); valentina.delledonne1@unicatt.it (V.D.D.); 2Neurology Unit, Fondazione Policlinico Universitario Agostino Gemelli IRCCS, 00168 Rome, Italy; camillo.marra@policlinicogemelli.it; 3Clinical Psychology Unit, Fondazione Policlinico Universitario Agostino Gemelli IRCCS, 00168 Rome, Italy; valentina.arcangeli@unicatt.it; 4Department of Woman, Children and Public Health, Catholic University of Sacred Heart, 00168 Rome, Italy

**Keywords:** language development, global cognitive development, screen time, children, technology

## Abstract

Screen time refers to the amount of time a child is exposed to a screen, that is, television, computer, smartphone, or any other digital medium. Prolonged screen time in the first years of life may affect a child’s cognitive abilities, especially language acquisition. A systematic review was conducted, following the PRISMA-P guidelines, with the aim to explore the available literature relating to the impact of screen time on children’s language development. This review identified 18 articles. The articles reviewed showed that prolonged screen time and exposure to screens in the first 2 years of life can negatively affect language development and communication skills, in terms of comprehension and vocabulary range. In addition, overexposure to screens in the early years can affect overall cognitive development, especially attention to environmental stimuli, social experiences, problem solving, and communication with others, e.g., the alternance of rhythms and roles in a conversation. In conclusion, our systematic review supports the idea that preschool screen time has negative effects on children’s cognitive and language development. Television seems to be the medium most detrimental to children’s skills, as it is used in a passive manner and is often characterised by language and content that do not suit the child’s processing mode. Future studies should increasingly focus on the digital media that children possess at an early age, such as mobile phones and tablets, and on how children relate to the online world, such as social networks.

## 1. Introduction

Technological progress is leading to an increase in the use of electronic devices and media, broadcast and interactive, not only among adults, but also among children. Broadcast media refers to television and movies, while interactive media includes, for example, social media and video games. Nowadays, preschoolers grow up in environments characterised by the Internet, computers, and video games on which they place much of their attention [1].

The use of electronic devices and media leads those who use them to spend time in front of a screen. The duration of time spent in front of a screen, whatever its nature (i.e., television, computer, cell phone, video games, or tablet), is defined as screen time [2]. It is possible to distinguish passive screen time and active screen time, depending on the use of electronic devices and the physical and cognitive engagement that the individual employs in the interaction with the screen. In passive screen time, there is no physical involvement or extensive expenditure of cognitive resources, and the individual does not interact with the screen, e.g., watching television. In active screen time, there is greater use of cognitive and physical resources, as there is interaction with what is happening on the screen, e.g., playing video games [3].

Prolonged use of screens can have detrimental effects on a child’s health and development [4]. Studies of young children reported significant associations between time spent on schematics and deficits in cognitive development in short-term memory skills and academic achievement in reading and math and language development [4,5,6].

The Council on Communications and Media of the American Academy of Paediatrics (AAP) issued Recommendations for Parents and Paediatricians to define a guide to the use of screens for children, advising against the use of screens for children in the first two years of life and indicating the need for parental proximity when using digital screens [7].

Also, The World Health Organization (WHO) has outlined guidelines recommending that children aged 0–2 should not be in front of a screen and that children aged 2 to 4 years should not be left passively watching a screen of television, smartphone or tablet for more than 1 h [8]. Studies by the Canadian Paediatric Society (CPS) have highlighted that only 15 % of preschoolers meet Canadian guidelines on limiting screen time to one hour per day [9], that television dominates screen time activities and is increasingly on the rise in children aged 3–5 years [10]. In the United States, media use rates among children aged 2 to 4 increased from 39 percent to 80 percent between 2011 and 2013 [11]. A British study found that about 51% of children aged 6 to 11 months, daily, are put in front of touch screens [12]. In Italy, there are little data on children’s media use. One survey showed that 20% of children use a smartphone for the first time during the first year of life. Eighty percent of children can use their parents’ smartphones between the ages of 3 and 5. Finally, parents use media to keep their children quiet between the first (30%) and second year (70%) of life [1,13].

The description of normotypic language development shows that children utter their first words between 10 and 15 months and acquire their first word combinations between 15 and 21 months. High levels of grammatical accuracy are evident by age 4 years [14]. Language acquisition is mediated by the left fronto-parietal cerebral network [15]. In particular, the crucial area for acquiring new words is the hippocampus [16]. The inferior parietal lobe and the left superior temporal gyrus allow the acquisition of auditory information [17]. The inferior frontal gyrus and left middle temporal cortex are involved in the processing and retrieval of memorised semantic information [18].

The literature shows that the link between screen time and language includes positive aspects and negative aspects [19]. Screen time can positively influence vocabulary expansion, children’s exposure to different cultural and linguistic values, and keep them safely occupied [20,21]. On the other hand, it should be emphasised that children begin to process and understand surrounding information from the age of 2, so they may have difficulties in processing data from screens. In addition, excessive exposure to screens may interfere with language development and reading and math skills. Finally, having television in the background, thus not as the main attentional focus activity, may have adverse effects on expressive language, executive functions, attention, and quality of play [2].

The screens to which children are exposed have changed over time. Exposure to television, computers, and video games has been joined in recent years by exposure to smartphones and touch screen technology, which is easier to use, even for pre-school children who lack reading and writing skills. As children at an increasingly early age are exposed to screens, it is necessary to understand the impact these have on their physical and mental health. Therefore, our systematic review aims to investigate the contributions in the literature over the last 20 years that examine the relationship between screen time, regarding both television and most advanced media, and language in children during the early years of development. Our purpose is not only to investigate the role that overexposure to screens plays in language development, but also to highlight what data have not yet been collected regarding the relationship between technology and the development of language skills in children, in order to pave the way for future studies that could fill these gaps. Since language acquisition is closely related to the other cognitive domains (attentional, perceptual, memory, and cognitive flexibility) and social interaction, our secondary goal was to gather information that was present in studies focusing on the relationship between language development and screen time, regarding its impact on the rest of cognitive and social skills and on brain maturation in the early developmental years as well. However, this collection does not claim to be an exhaustive overview of the relationship between screen time and non-linguistic skills for which we refer you to a specific in-depth study.

## 2. Material and Methods

The current study was conducted following the PRISMA-P (Preferred Reporting Items for Systematic Review and Meta-Analysis Protocols) guidelines [22]. Our systematic review was not registered in the PROSPERO Database.

### 2.1. Data Sources and Search Strategy

To gather relevant literature, an online search was performed on PubMed/Medline and Scopus databases using the following research string: “(screen time) AND (LANGUAGE ‘ACQUISITION’)”. The search was conducted in November 2023. Our comprehensive search strategy was designed to identify relevant studies related to the impact of screen time on language acquisition in early development. We also collected data in the selected studies about the impact of screen time on other non-linguistic domains, social skills, and brain maturation. This information was incorporated into our work, although we did not actively search for such data because they can help us understand the factors that contribute to language acquisition and the association of difficulties in these domains exacerbated by screen time.

### 2.2. Study Selection

Abstract screening for eligibility was independently carried out by two authors (VDD and VM). In the cases of any discrepancies, a third, senior author (DPRC) was consulted for consensus. As part of the inclusion and exclusion criteria, studies published more than 20 years ago were precluded from consideration. The absence of clear information on the impact of screen time on language in the child population present in the title and abstract was applied as an exclusion criterion at the screening stage. Papers were retained if there was insufficient information to exclude them. The remaining articles were reviewed. Only English-language articles and peer-reviewed publications were considered. Additionally, the following exclusion criteria were applied: papers out-of-scope with a focus not centered on language development, papers involving a population with diseases not covered by our study, reviews and/or systematic reviews, letters to the editor, case reports. Furthermore, a systematic abstract screening of the references (forward search) was conducted to identify any additional relevant records.

## 3. Results

The study selection process is illustrated in Figure 1. The search of the literature yielded a total of 278 results. Duplicate records were then removed (*n* = 0). In total, 260 records were excluded after abstract or full text reading: 182 records were identified as ‘out of scope’ of our review and 28 records were excluded as review or meta-analysis; in addition, a further 50 records were excluded because they contained the concept of screen time applied to diseases, not the subject of our study.

Therefore, 18 studies were included in the review (Figure 1). The main findings of the studies, classified per pathology and detailing study design, number of cases, and major observations, were summarised in Table 1.

### 3.1. Characteristic of the Included Studies

As shown in Table 1, we selected 18 studies examining the influence of screen time on language development in children. A total of 32.274 pre-school children between 0 and 5 years and 414 school-age children between 6 and 7 years were included in the selected studies.

Eleven of the included studies investigated both active and passive screen time, seven studies explored only passive screen time, and only one study fully active screen time.

In total, 5 of the included studies exclusively investigated language development, 11 studies assessed language development together with the overall development of children including cognitive, motor, and behavioural levels, 2 studies also included an assessment of a parent–child interaction, and 1 focused on ASD symptoms.

A total of 9 studies were from North America, 4 from Asia, 3 from Europe, 1 from Australia, and 1 from South America. Nine were cross-sectional, five were longitudinal, four were prospective, and two were pilot studies. 

### 3.2. Screen Time and Language Development

From the data obtained both from direct assessment of children’s level of language development and from questionnaires given to parents reporting data on their children’s screen use, seven studies showed that prolonged passive screen time, especially television, influences a child’s verbal activity, reducing it [24,25,26,27,32,37,38]. Watching television without guidance reduced verbal activity and increased the risk of developing a delay in language acquisition 8.47 times more than children who interfaced with their parents during television programming [24]. Children who watched television for more than 4 h a day had a higher risk of developing delayed language acquisition, and 70% of children with delayed language acquisition had television in their rooms [27]. In addition, a passive activity such as watching television seemed to lead to a reduction in vocalisations in children aged 2 to 48 months and a difficulty in implementing communicative shifts with the intention of having relational contact with caregivers. At the same time, a reduction in the verbal activity of parents emerged as well, who produce 500 to 1000 fewer words in interactions with their children [25]. Still, other studies we selected showed how moments of family interaction are important for language development and how these can be undermined by television use. For example, some authors pointed out that watching television during mealtimes is negatively associated with a reduction in expressive language and verbal interaction patterns in children aged 2 to 5 years [32,38].

Direct assessment of the children showed that difficulties in verbal interaction with caregivers and the surrounding environment due to the prolonged screen time can also lead to a lexically deficient acquired language, especially the reduced expansion of one’s expressive and communicative vocabulary [26,31,32,38,39]. Watching too much television has potential adverse effects related to deficient verbal expression. Prolonged screen time, therefore, seemed to cause a deficient expressive vocabulary and poor phonological processing [32,38]. This occurs especially in cases of screen time of more than 4 h per day in children aged 2 to 4 years, as evidenced by direct assessments of children’s language [32,38,39]. This can also be evidenced by alterations in the cerebral white matter in the early years of development, especially in the areas deputed to language processing, such as in the arcuate fasciculus that supports the semantic processing of words [31]. Other studies we selected, however, did not support such a correlation between screen time and lexical difficulty [33,34,35,40]. There are authors who claimed that there is no significant relationship between screen time and deficient language acquisition or expressive vocabulary efficiency [33,35]. Others emphasise the need to distinguish the type of screen used. Based on the results of parent-report questionnaires, smartphones and tablets, if used in an active manner and involving the family, can be a useful learning tool, which is helpful in increasing expressive performance [34]. Watching educational programmes on television can also correlate positively with expressive vocabulary [40].

Finally, one author emphasised how passively watching programmes on television that differ from one’s own linguistic and cultural background, without guidance explaining what one is watching, leads to language acquisition difficulties in children at the grammatical level of construction, comprehension, and order of the sentences heard [27].

### 3.3. Screen Time and Global Development of Children

Five of the selected studies investigated the relationship between screen time and developmental milestones in children further than language. Direct assessment of the children showed that those between 12 and 35 months exposed to television for 2 h a day had about 4 times the chance of developing a cognitive developmental delay and 3.7 times the risk of developing a motor developmental delay compared to those exposed less frequently [26]. Moreover, children with delays in critical developmental milestones tended to have higher screen time than children with normotypic development [26].

Questionnaires completed by parents also showed that children exposed to longer screen times at 24 and 36 months were more likely to perform worse at developmental screenings at 36 months and 60 months, respectively [29]. They showed that their sample of 24-, 36-, and 60-month-old children have roughly 2.4, 3.6, and 1.6 h of screen time per day [29].

Daily screen time exposure >4 h in 1-year-old children was associated with developmental delays reported by parents at both 2 and 4 years of age only in the domains of communication and problem-solving [39].

From the data obtained both from direct assessment of the children and from parent report questionnaires, it was found that more screen time was associated with lower overall cognitive development at 3.5 years and 5.5 years, especially in fine motor, language, and self-regulation skills, but better non-verbal reasoning at 3.5 years. Watching television during meals was negatively associated with general cognitive development only at 3.5 years [38]. The authors also showed that the associations between screen time and cognitive development are nonlinear at 2 years and linear at 3.5 and 5.5 years.

Preschool children around 5 years of age with a screen time >1 h had a greater vulnerability, reported by parents, in cognitive (81%), communicative (60%), social (60%), physical (41%), and emotional maturity (29%) development than children with a screen time <1 h [36].

Seven of the selected articles explored the association between screen time and cognitive and behavioural abilities. Regarding attentional skills, based on the results of parent-report questionnaires, screen time on television before the age of 3 years related to entertainment content appears to be associated with attentional problems at 5 years, with no difference between violent and non-violent content [23].

Direct assessment of the children showed that pre-schoolers between 36 and 60 months who exceed the daily screen time by 1 h were about three times more likely to have a worse working memory than those who do not exceed this limit [35].

Also, in 3-year-old children, total screen time and specific video/show/movie viewing were negatively correlated with working memory either from a direct evaluation or from parent-report questionnaires. In addition, co-use of traditional and mobile screen devices was negatively correlated with self-control, whereas screen time related to educational content has a positive correlation with inhibitory control [40].

In extremely premature school-age children (between 6 and 7 years) having a daily screen time >2 h was associated, both on the basis of direct evaluation and questionnaires filled in by parents, with lower IQ levels and deficits in executive (including metacognition) and attentional functions than those <2 h. Having a television or computer in the bedroom was also associated with increased problems with inhibition, hyperactivity, and impulsivity [33].

The amount of television/DVD/video screen time in children aged 2–5 years was inversely associated with social compliance skills, reported by parents, while the amount of outdoor play time was positively associated with expressive and social compliance skills [28].

Results of parent report questionnaires and direct assessment of the children showed that screen time >1 h in children aged 2 to 5 years was significantly associated with lower cognitive and socio-emotional abilities but not with lower motor skills or hand fine motor coordination. A screen time >5 h per day in mothers was associated with higher media use in children, and higher levels of parent–child interaction were associated with better cognitive and social–emotional abilities in children [37].

Furthermore, exposure of >4 h daily to television/video and little time in interactive play between caregiver and child were significantly associated with more frequent Autism Spectrum Disorder (ASD)-like symptoms, reported by parents, at 12 months but not at 18 months [30].

## 4. Discussion

### 4.1. Television and Language Development: How Television Affects Language Acquisition in Children

Zimmerman & Christakis [23] highlighted how educational, nonviolent entertainment, and violent entertainment television programmes may affect the language development of children under the age of 3 years differently. Television is intended for an adult audience and can be a distraction for parent–child communication. When using digital screens, proximity to the parent is necessary. The caregiver must be close to the child and pay attention to the type of content viewed, to explain passages that may be distant from reality on an informational level, and to prevent the child from being alone in interacting with the media [7]. Adult entertainment programmes, especially the more explicit and violent ones, are loud and use language more like adults in rhythm and pronunciation. This language is very different from maternal language, termed “motherese” by neuroscientists [41,42,43]. Motherese language allows the mother to get in touch with the children’s needs and to modulate her actions based on the child’s needs. In addition, motherese facilitates the natural alternation between the two communicative subjects, i.e., the language giver and the language receiver, and allow the young child to acquire language in a facilitated social context [44,45]. Educational programmes, on the other hand, more closely match the communicative timing of maternal language, imitating its rhythm, pronunciation, and bringing the child closer to language acquisition in a more natural way [46]. Ling et al. [26] highlighted that children younger than 2 years old are also exposed to television for times longer than one hour per day. Children over the age of 2 spent at least more than 2 h a day in front of the television. The higher the television screen time, the more this negatively affected language development. Schwarzer et al. [37] showed that high screen time negatively affects language development between the ages of 2 and 5 years, especially prolonged television viewing. Prolonged maternal television screen time, on the other hand, did not affect children’s language development, but was significantly associated with the amount of time children spend watching television. Furthermore, television used as a background leads to distortions in the deciphering of sounds at a phonological and syntactic level, and this causes limitations in language learning and vocabulary expansion, culminating in poor language at an expressive level [32,38]. Takahashi et al. [39] showed that greater screen exposure at age 1 leads to more enduring deficits thereafter, especially in communication. In fact, children who were exposed to the screen at age 1 for both 1 to 2 h, 2 to 4 h, and more of 4 h manifested lasting deficits in language compared with children exposed less than 1 h per day, and these deficits tended to persist over time.

### 4.2. Passive Screen Time vs. Active Screen Time: Do All Media Have the Same Effect on Child Language Development?

Nobre et al. [34] investigated the incidence of prolonged screen time on children aged 24 to 42 months, a crucial period of development. In this study, children were exposed more to television, with less exposure to interactive media, such as smartphones and tablets. Only a few children in this age group used video games. This study confirms the delay in language acquisition due to prolonged exposure to television but emphasises how important it is to specify the media used, as interactive media can contribute to children’s vocabulary increase, with a significant contribution to literacy skills. The authors explained this result by serving data in the literature that distinguish between passive and active screen use and the ability to make use of the parental figure during screen use, as active screen time and interaction with the external environment promotes lexical development, which passive television viewing does not.

Data in the literature [47,48,49], however, underscore the importance of considering certain strengths and weaknesses of interactive media. For example, it appears necessary for an adult to be present as a mediator between the digital media and the child and for the child to use it to learn or have fun, not out of boredom or as a means for the child to do something while the adult is busy. Only an interaction with the environment can enable the acquisition and enhancement of language development. In addition, a distinction must also be made according to the type of interactive medium one meets. In fact, Reich et al. [50] pointed out that eBooks can be distracting or incapable of attentional focus in young children because of their sounds and animations that require great attentional selection and processing skills.

Sweetser et al. [3] showed that children between the ages of 2 and 5 years have prolonged screen time that exceeds paediatric recommendations, especially for television and DVD viewing, compared to video game and computer use. The authors emphasise that video game and PC use should be considered differently from time spent watching television or a DVD because of the psychological and physical implications. Some games, for example, are designed to promote physical well-being by simulating running, jumping, or other bodily movements that must be adopted to pass the levels offered by the game [51]. Others, on the other hand, require academic skills to be won and allow community play for their success, fostering social skills [52]. Computer use can increase attention skills, visuomotor coordination, and alertness in processing sensory stimuli [53,54], while prolonged viewing of television or DVDs can adversely affect short-term memory, attention, language development, and vocabulary range [6,55,56]. So, the authors point out that a distinction must also be made between the type of media used, as well as by the time of exposure. In addition, the content viewed or with which the child comes into contact is important. For example, violent video games should not be used by young children [3].

### 4.3. Do All Media Affect Language Development? Discordant Studies on the Negative Relationship between Screen Time and Language Acquisition

Vohr et al. [33] showed that prolonged screen time in children born prematurely can have negative effects on global cognitive functions but not on the sphere of language. They thus emphasise how expressive language may not be as sensitive to television exposure as receptive language in early life.

In 2022, Zhang et al. [35] examined the association between screen time and the development of cognitive domains in pre-school children. The amount of time spent on devices, regardless of medium, was not found to be related to expressive vocabulary, but to difficulties related to working memory. The authors hypothesise that there is no association between screen time and vocabulary in their study, compared with others in the literature, because expressive language is not as susceptible to exposure to television as receptive language in the early years of life since it is a rather small vocabulary in emission compared with children’s comprehension faculties.

### 4.4. Child–Parent Interaction: How Caregivers Can Mediate the Effects of Screen Time on Language Development

Interaction with the caregiver can mediate the negative effects of screen time. Shared activities with parents can foster language development, such as reading a book together [7]. Christakis et al. [25] showed that when children between 2 and 48 months are left alone with the television on or if there is no presence of adult interaction, the television can lead to reduced vocalisations by the children, less interest from the parents with a reduction in their words addressed to the child as well, and consequently, poor shifts in conversation between parents and children. So, parents tended to engage children less when the television is on, and this spills over negatively not only into language development, but also into attention and cognitive delays. The authors pointed out that it is important to consider media exposure not only in terms of the amount of use of the device or the content offered by the device, but also by defining context, that is, how a system is used in the surrounding environment. So, the mode of interaction with parents and the manner and number of words the parent utters toward the child while watching television programmes is as important in this age group as what is being watched. A guide enables children to have an explanation of what they are seeing, especially if the content being viewed is different from their cultural background and cannot be understood by passive learning alone [27]. Ling et al. [26] pointed out that interaction with the caregiver can mediate the impact of screen time on child development. Interaction with the parent turned out to be necessary for the child to expand his vocabulary and enrich himself with elements of his environment. It has been highlighted how children with mothers who spend a lot of time away from home at work and with a low level of schooling spend more time in front of the television, especially if they are only children. This also happens when children are left with grandparents or nannies, due to the little time parents have to spend with them. These figures may be less strict with children, allowing them to use computer media to distract themselves but aggravating their cognitive and physical development [57,58]. Madigan et al. [29] pointed out how prolonged screen time in young children can interfere with the relationship with the parent, limiting the opportunity for verbal and nonverbal social exchanges necessary for growth. Schwarzer et al. [37] highlighted how a parent–child interaction can help maximise language development. In fact, prolonged screen time may limit children’s opportunities in communication with others, interaction with the environment, and play, leading to hyperactivity and inattention [28,55,59]. The results of the study suggest, therefore, that 1 h of media use in preschool children is associated with worsening language, while a parent–child interaction is found to promote expressive development [8]. Yang et al. [38] found how having the television on during family meals affects expressive language development in 2-year-old children. Prolonged screen time may affect social activities and interactions that indirectly harm language development. The authors described children’s first year of life as crucial to their development and maturation, so excessive screen time can lead to difficulties in benefiting from other play activities and social interactions, indirectly preventing language acquisition.

This agrees with the results of Martinot’s court study, which showed that television on during family meals negatively affects language skills from ages 2 to 6 [32]. In the present study, in fact, it was seen that more than an hour-long television screen time per day and watching television during meals affect important milestones in language acquisition related to vocabulary, comprehension, and production of gestures with communicative intent. This is because when television interferes with the quality and quantity of a parent–child interaction, an essential stage of language acquisition in childhood is altered. In addition, noise from television can create confusion in children in deciphering the phonological sounds and syntactic rules of their surroundings, limiting verbal comprehension and expression [60,61]. Rai et al. [40] showed that in 3-year-old pre-school children, prolonged screen time is for watching videos (shows, movies, or social media videos), during which there is a reduced interaction with parents. In contrast, less prolonged screen time occurs for the use of digital games, but this brings a better quality of interactions with parents. Increased screen time for watching videos and reduced interaction with their surroundings is negatively associated with the development of expressive vocabulary. This is because it appears that parents use electronic devices to keep children occupied while they do other activities, so children use these systems in a passive mode, which affects their environmental communication skills [62,63].

### 4.5. Impact of Screen Time on Children with Delayed Language Development

Chonchaiya and Pruksananonda [24] showed that children with a language developmental delay tend to watch television as early as 10 months before the emission of their first meaningful word. Those without a language developmental delay started watching television after they utter their first meaningful word. Children who start watching television more than two hours a day and at an age younger than 12 months had a six times greater risk of developing a language developmental delay than children who do not engage in this behaviour. In addition, the authors highlighted that 60% of children with delayed language development tend to watch television alone, while those with normotypic language development had more interaction with their caregivers. Language development involves social interaction with one’s reference figures and a rich environment full of communicative stimuli. Television, even if used as a background and not as a primary activity, tends to distract the child from his surroundings and play and negatively affect communicative interaction with other family members [64,65]. Chonchaiya and Pruksananonda [24] pointed out that there seems to be an association between prolonged screen time at an early age for television and delay in language development, although this is not a direct cause-and-effect relationship. Television is a complex medium that requires brain maturation and complex cognitive skills that typically develop after the age of 2 or 3 years [66,67].

Perdana et al. [27] showed that most children with both normal language development and delayed language acquisition are exposed to television before the age of 2. Children with delayed language development tended to watch television for longer. This finding was not significantly associated with delayed language acquisition, but the authors pointed out that, potentially, children with a language delay are left alone to watch television without an adult to interact with them. Children who watch television more than 4 h a day had a higher risk of developing delayed language acquisition. Having television in the room was not found to be a significant variable in the development of delayed language acquisition. Children who watch television in a language other than their native language, without their parents beside them, had higher risk of developing a language delay. The authors pointed out that in young children, watching a programme in another language without guidance can lead to confusion and difficulty in comprehension. This is because without a guide to interact with them and explain what is going on, children may have difficulty understanding what is happening on the screen and confusion in internalising what they are hearing since they are not words they hear every day [68]. Listening to a language other than their own for children is passive learning, which inhibits the more functional learning made up of syntactic and grammatical rules under construction [69,70]. The authors pointed out that television per se is not detrimental to language development. What is more problematic is the lack of the stimuli. In fact, the children at greatest risk of developing delayed language acquisition are those who watch more television in lieu of interacting with their parents and surroundings [71,72].

### 4.6. Brain and Prolonged Screen Time: Structural Alterations in White Matter and Influence on Language Acquisition

Hutton et al. [31] described the structural neurobiological correlates related to screen time in preschool children who were not yet in school and not reading independently. The authors pointed out that prolonged screen time is associated with a deficient microstructural integrity of the brain’s white matter tracts that support language development. To assess the microstructural integrity of white matter, they used Diffusion Tensor Imaging (DTI) as a technique and the following parameters [73]: fractional anisotropy (FA—associated with the organisation of white matter into parallel bundles) and radial diffusivity (RD—inversely associated with the degree of myelination of parallel bundles of white matter). Normally, in response to constructive environmental stimuli, such as exposure to language, children show increases in AF and decreases in RD [74,75,76]. Increased screen time is associated with decreased FA and increased RD in the arcuate fascicle, which connects receptive (Wernicke) and expressive (Broca) brain areas associated with phonological processing and vocabulary [77,78,79]. In fact, such microstructural alteration lead to lower results in tests assessing expressive vocabulary and phonological processing [31]. Similar associations are also observed for the uncinate fasciculus and inferior longitudinal fasciculus that support other aspects of language, such as semantic processing. In addition, increased screen time appears to be associated with decreased AF in the uncinate fasciculus and fronto-occipital fasciculus in the left hemisphere, which are involved in naming words and objects in a rapid and automated manner [75,79]. Finally, reduced FA have also been described in the inferior thalamic tract and minor forceps, which are associated with the speed of information processing and working memory that support language tasks. So, early and long-term exposure to screens lead to a lower pre-reading ability in young children and adversely affect language development [80,81].

### 4.7. Association between Global Cognitive Development and Screen Time across Ages and Developmental Domains

The first five years of life are a crucial time in children’s development, critical to their growth and maturation, and excessive exposure to screen time can hinder proper development and school readiness. The results of the studies included in the review showed an association between an earlier exposure of children before the age of 3 to television and an increased likelihood of developing a delay in cognitive and motor development and not reaching certain developmental milestones [26,34]. Even preschool children of about 5 years of age who exceed the recommended amount of daily screen exposure of 1 h appeared to show an increased vulnerability in the development of social and cognitive skills, emotional and social maturity, and physical and mental well-being [29]. Potentially, these children spend less time on activities that can enrich cognitive and social development and may miss important opportunities to practise motor, interpersonal, and communication skills. More time in front of screens can interfere with health-promoting experiences such as physical activity, proper sleep hygiene, and social contacts with peers and family [26,32,37].

This finding is in line with existing studies that have shown that excessive daily screen use has a deleterious effect on cognitive development [29,82,83,84,85]. Prolonged screen time during a critical period of growth and maturation can undermine optimal development by limiting the practice of interpersonal and communication skills [86], thus slowing social and language development [60,87].

However, the association between cognitive development and screen time appears to vary between ages and developmental domains. Takahashi et al. [39] showed a negative association exclusively between prolonged early exposure to screens in 1-year-old children and the developmental domains of communication and problem-solving at 2 and 4 years of age. The authors also showed that this negative association decreased over time. Same for social development and fine motor skills: these skills were shown to be low in 2-year-olds with high screen exposure, but the problem disappeared by age 4. Thus, the authors showed how time spent in front of the screen affects some areas of development in fact, but not others and that not all associations persist.

The study by Yang et al. [38] found a negative effect of screen time on global cognitive development only at 3 and 5 years. On the other hand, a positive effect on non-verbal reasoning skills at 3.5 years and between an intermediate screen time at 2 years and cognitive development is shown. Children not exposed to the screen for an excessive amount of time may be exposed to screens under parental supervision and be more likely to watch high quality content (e.g., educational programmes or some training video games), which may promote their language and general cognitive development.

It is therefore possible that not all children are equally affected by screen time but that there are factors that mitigate the negative effects of the screen on a child’s development. It is appropriate to consider the developmental domains separately when investigating the effects of screen time.

### 4.8. Impact of Screen Time on Attentional and Executive Functions

Attention skills and executive functions also seem to be affected by screen time in children. The type of content of television programmes to which children are exposed before the age of 3 appeared to influence the onset of attentional problems 5 years later. Only entertainment programmes and not educational programmes seemed to be associated with the onset of attentional disorders. Entertainment programmes use an adult language in rhythm and pronunciation, which is different from the mother’s language and are not in tune with the child’s growing abilities [88].

Vohr et al. [33] reported that excessive screen exposure in early school years in preterm children is associated with an increased risk of attentional and executive problems. Having a television or computer in the bedroom was also associated with behavioural problems of inhibition, hyperactivity, and impulsiveness. This association between increased screen time or the presence of a television/computer in the bedroom and increased rates of attention deficit/hyperactivity disorder, attention problems, and hyperactivity is also confirmed by other authors [89,90,91]. This impact may be amplified in extremely preterm children [33].

Total screen time was also found to be unfavourably associated with working memory in children up to the age of 5 [36] and specifically video/show/movie viewing [40]. Furthermore, preschoolers who complied with the 1 h screen time recommendations were more likely to have a better working memory than those who did not [36], and screen exposure time related to educational content had a positive correlation with inhibitory control [35]. The authors pointed out that adult entertainment programmes with a faster and more articulate cadence and pronunciation affected working memory abilities, especially when placed as a background to daily routines. Prolonged screen time could distract from more appropriate environmental experiences (such as interacting with family or reading) and inhibit the ability to attend to environmental stimuli [92]. One possible explanation is that rapid image changes on the screen tend to hamper children’s sensory processing and attention skills, leading to difficulties in filtering out relevant stimuli and, thus, impairing working memory performance in the case of excessive exposure [93,94].

On the other hand, intermediate exposure to educational content, i.e., under parental supervision, appeared to be a protective factor with regard to working memory, consistent with other studies showing that high-quality content (e.g., adult-directed, educational, slow-paced) is less unfavourable for executive functions in pre-school children [95].

However, Rai et al. [40] reported that the shared use of traditional and mobile screen devices with parents is negatively correlated with self-control in 3-year-old children. This can be explained by the fact that watching videos was the most common type of screen time and that watching videos had the lowest quality of parent–child interactions. Thus, it appears that only high-quality interactions between parents and children can mitigate some of the negative impacts of screen time through the joint use of devices by providing opportunities for a parent–child interaction, support, and feedback [96,97].

### 4.9. Screen Time and Development of Social Skills

Schwarzer et al. [37] also showed that higher levels of a parent–child interaction are associated with better cognitive and socio-emotional skills in pre-school children and that prolonged screen use by mothers is not directly related to child development, but to the average use of screens by children. It therefore seems important to sensitive and educated caregivers to mediate and role-model their children’s media habits by moderating their own media habits and providing better parent–child interactions.

High passive screen time leads to a reduction in social and interpersonal interaction skills in pre-school children. Hinkley et al. [28] showed that early excessive exposure to television between the ages of 2 and 5 is inversely associated with social skills as opposed to time spent playing outside the home. Moreover, using computers, electronic games, or laptops had no negative association with the social skills scale. This result is consistent with previous studies reporting inverse associations between TV viewing and prosocial behaviour [98], cooperation, assertiveness, self-control, and social skills in general [99]. This depends on how the instruments are used. For example, television leads to being a passive consumer of content, whereas a computer game leads to greater physical, cognitive, and social engagement, as it may require the cooperation of friends or parents [100].

Exposure to the screen, especially in passive forms, can limit children’s opportunities to interact with others, and even the mere presence of a television set in the background has been shown to diminish a parent–child interaction [60]. On the contrary, play and physical activity provide opportunities for interaction, conversation, cooperation, and conflict management, allowing children to learn important social skills [101].

Heffler et al. [30] showed how excessive exposure to television and insufficient exposure to interactive play is associated with more frequent ASD symptoms at 12 months, confirming that increased screen exposure in young children interferes with social learning, and demonstrating that it is associated with altered brain processing and could theoretically promote hyperconnectivity of the visual brain. However, the authors pointed out that children predisposed to ASD may have a preference for screens or that screens may be used by parents as a method of calming those children with self-regulation problems. Finally, the authors pointed out that daily play interactions between parents and children are associated with fewer ASD symptoms. Although this association was modest, it should be noted that the interaction with the caregiver can mitigate the ASD symptomatology.

## 5. Practical Fallouts

Our systematic review can support the analysis of the impact of screen time on preschool children. Selected studies show that prolonged screen time in the under-2 age group and in the 2-to-4 age group can bring issues that spill over into the child’s entire development: from global cognitive development to language acquisition to the ability to interact with others. Television seems to be the medium most damaging to childhood skills, as it is used in a passive mode and is a medium of understanding for an adult audience, since language and content do not fit the child processing mode. Future studies will increasingly need to focus on the digital media that children possess early, such as cell phones and tablets, and how the child relates to the online world, such as social networks, since the age of enrolment in actual reality violates the established age limit and is increasingly earlier, just as social content made by parents with their children and disseminated on the Internet is not uncommon. To do this, there should be synergistic work between different child health professionals (paediatricians, psychologists, and child neuropsychiatrists) and parents. The central goal of this alliance should be to help parents understand the effects of prolonged screen time on their children’s development and to provide them with behavioural strategies to adopt in order to establish a healthy routine of media use by children [102]. In constructing the routine, it is helpful to advise parents to alternate between screen time and off-screen time and to never leave the child, especially at an early age group, alone in front of a screen, but to stand next to their child and interact together with questions and explanations about what they have just viewed [103]. It seems, therefore, crucial to promote research that expands the data on screen time, especially for active screen time related to more modern media, to implement early prevention interventions and to foster the optimal development and physical and mental well-being of the child and the entire family unit in relating to them.

## 6. Conclusions

From the results of our review, it seems that monitoring the use of digital devices in preschool children is necessary to safeguard and protect their development. It seems crucial to consider the ways in which screen time can affect the psychophysical development from the child in the early years of life so essential to his or her development—from structural changes in the brain’s white matter, to the development of language, to the emission of social behaviours in interacting with the surrounding environment. Evidence from the literature emphasises the importance of investing in the health of young people from an early age to restrict excessive screen time and its consequences throughout development, such as a delay in cognitive, language, and social development; isolation and a lack of interaction with the circumstance environment; a sedentary lifestyle and lack of movement, resulting in being overweight or obese; and mood issues.

Finally, it should be emphasised that new technologies are advancing rapidly in all areas of social, working, and everyday life. Children are confronted with different media and information media, at pre-school and school level, not only as a means of entertainment, but also as a teaching and learning tool. Our review shows that studies in the literature mainly focus on passive screen time, leaving a gap in the analysis of active screen time. Further studies are needed to highlight the effects of active screen time on the neuro-biological, cognitive, and psychological level in the development of children’s global abilities. In this way, the effects of active screen time and any differences with the reported effects of passive screen time can be described qualitatively and quantitatively.

## Figures and Tables

**Figure 1 brainsci-14-00027-f001:**
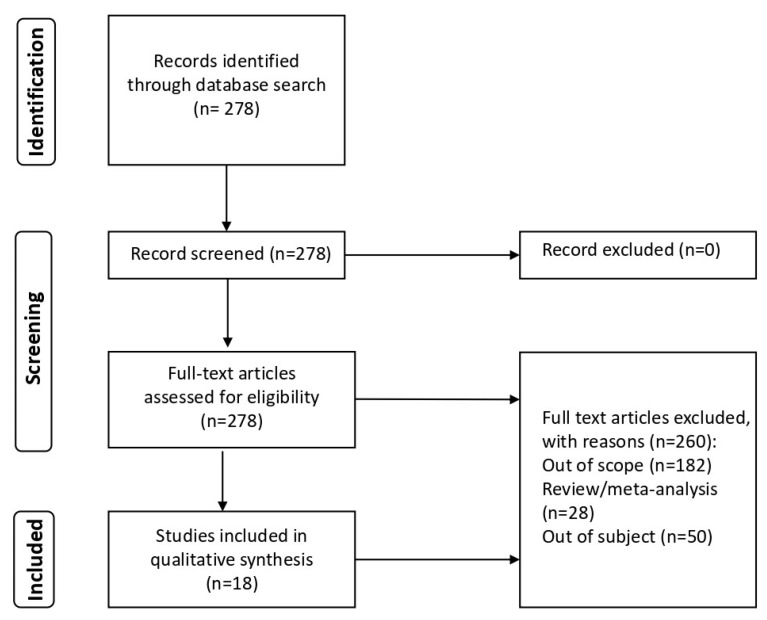
Flowchart of search strategy and selection of reports.

**Table 1 brainsci-14-00027-t001:** Characteristics of included studies.

	Author	Year	Study Design	Number of Cases	Materials	Main Findings
1	Zimmerman F.J. & Christakis D.A. [23]	2007	Longitudinal study	560 children’s parents aged 0 to 35 months. 407 children’s parents aged 4 to 5 years.	Behaviour Problems Index (BPI) the Child Development Supplement (CDS)	Early television viewing is associated with subsequent attention problems specifically for non-educational content and viewing before the age of 3.
2	Chonchaiya W. & Pruksananonda C. [24]	2008	Pilot study of a case-control study	56 patients with language delay and 110 normal children, aged 15–48 months.	Denver-II	There is a relationship between early onset (at less than 12 months of age) and high frequency of television viewing (>2 h per day) and language delay.
3	Christakis D.A. et al. [25]	2009	Prospective, population-based observational study	329 children from 2 to 48 months	LENA Software (LENA Foundation, Boulder, Colorado): adult word counts, child vocalisations, and child conversational turns	Sound television is associated with less exposure to discernible human speech of adults and less speech of children, contributing to the delay in language development.
4	Ling-Yi L. et al. [26]	2015	Cross-sectional study	75 children frequently exposed to television and 75 children not frequently exposed to television, between 15 and 35 months of age.	Bayley Scales of Infant Development—second edition (BSID-II) Peabody Developmental Motor Scales-second edition (PDSM-2)	Children under 35 months of age exposed to television for 2 h a day are more likely to develop delayed language, cognitive, and motor development.
5	Perdana S.A. et al. [27]	2017	Cross-sectional study	84 children from 18 month to 3 years.	Developmental Pre-Screening Questionnaire (Kuesioner Pra Skrinning Perkembangan or KPSP) Early Language Milestone (ELM Scale-2)	Children who watch television more than 4 h a day and who watch television programmes in both English and their native language (Indonesian) have a higher risk of developing a language delay.
6	Hinkley T. et al. [28]	2018	Cross-sectional study	958 children from 3 to 5 years	Adaptive Social Behaviour Inventory—ASBI	Television/DVD/video viewing may be negatively associated, and outdoor play favourably associated, with the social skills of pre-school children.
7	Madigan S. et al. [29]	2019	Longitudinal cohort study	2441 mothers and children. Data were available when children were aged 24, 36, and 60 months.	Ages and Stages Questionnaire, Third Edition (ASQ-3)	Higher levels of time spent in front of the screen in 24- and 36-month-old children are associated with a delay in reaching the developmental milestones of children at 36 and 60 months, respectively.
8	Heffler K.F. et al. [30]	2020	Prospective cohort study	2152 children at 12 and 18 months of age	Modified Checklist for Autism in Toddlers	Excessive daily exposure to television/video and little interactive play time between caregiver and child is significantly associated with more frequent ASD symptoms.
9	Hutton J.S. et al. [31]	2020	Cross-sectional study	69 children between the ages of 3 and 5 years old	ScreenQ survey Expressive Vocabulary Test, Second Edition (EVT-2) Comprehensive Test of Phonological Processing, Second Edition (CTOPP-2) Get Ready to Read! (GRTR) Diffusion Tensor Imaging (DTI)	Excessive screen use is associated with worse results on cognitive assessments and lower measures of the microstructural organisation and myelinisation of brain white matter tracts that support language and literacy skills.
10	Martinot P. et al. [32]	2021	Cross-sectional and longitudinal analyses	1562 children aged 2 to 5–6 years old	MacArthur–Bates Communicative Development Inventory (CDI) the Evaluation du Langage Oral de L’enfant Aphasique and A Developmental NEuroPSYchological Assessment batteries	Screen time in 2-year-old children is negatively associated with language development; exposure to television during family meals is negatively associated with language test scores at all ages.
11	Vohr B.R. et al. [33]	2021	Prospective cohort study	414 children from 6 years 4 months to 7 years 2 months	Wechsler Intelligence Scale for Children—IV the Behaviour Rating Inventory of Executive Function the Developmental Neuropsychological Assessment (NEPSY II) the Conners 3rd Edition—Parent Short Form the Social Communication Questionnaire	High screen time is negatively associated with language, cognitive, behavioural, and executive function outcomes in extremely preterm children (EPT) at the age of 6–7 years.
12	Nobre J.R. et al. [34]	2021	Cross-sectional, descriptive, and exploratory study	172 children aged 24 to 42 months and 15 days	Bayley III test Family Environment Resource Inventory—FERI	Daily exposure > 2 h to television appears to be associated with delayed language, difficulties in social interactions, sedentary lifestyle, and low stimulation for creative thinking.
13	Zhang Z. et al. [35]	2022	Cross-sectional study	97 preschoolers (36 to 60 months)	Early Years Toolbox-YET Expressive Vocabulary Mr Ant’ task	Children under the age of 60 months who interact with the screen for more than one hour a day have a higher risk of having a worse working memory than those who do not exceed this limit.
14	Kerai S. et al. [36]	2022	Cross-sectional study	2983 children, the mean age of children was 5.2	Childhood Experiences Questionnaire—CHEQ Early Development Instrument—EDI	Children with a screen time of more than one hour per day, compared with those with a lower screen time, were more likely to develop vulnerabilities in the following domains: Social skills, emotional maturity, physical and mental well-being, language, and cognitive development.
15	Schwarzer C. et al. [37]	2022	Cross-sectional cohort study	296 healthy 2–5-year-old preschoolers and 224 mothers	German Health Interview and Examination Survey for Children and Adolescents (KiGGS)—Adaptation; Questionnaire on preschool-aged children’s activities in the family (AKFRA) ET 6-6-R: Development Test 6 Months to 6 Years—Revision	Daily screen time > 1 h in children and lower levels of parent–child interaction are negatively associated with language, cognitive, and socio-emotional abilities.
16	Yang S. et al. [38]	2023	Longitudinal cohort study	13.763 children aged 2–5.5 years old	MacArthur–Bates Communicative Development Inventory—MB Picture Similarities subtest from the British Ability Scales—PS Child Development Inventory—CDI	Family mealtime use of television in 2-year-olds is negatively associated with expressive language and general cognitive development.
17	Takahashi et al. [39]	2023	Prospective cohort study	7097 children from 2 to 4 years	Ages & Stages Questionnaires—Third Edition (ASQ-3)	Daily screen time > 4 h in children is associated with delays in the development of communication and problem solving skills.
18	Rai J. et al. [40]	2023	Cross-sectional study	44 children aged 3 years and their parents	word span test HTKS test iPad-based Early Years Toolkit modified snack delay test Parent–Child Interaction System (PARCHISY)	Screen use is not totally negative. Total time spent on the screen and specific video/show/movie viewing are negatively correlated with working memory and co-use of mobile devices is negatively correlated with self-control, while viewing educational content has a positive correlation with children’s inhibitory control.

## Data Availability

Not applicable.

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
