# Peer review of "The Relationship between Language and Technology: How Screen Time Affects Language Development in Early Life—A Systematic Review"

_brainsci, 2023, doi:10.3390/brainsci14010027_

Round 1
Reviewer 1 Report
Comments and Suggestions for Authors
This article is a meta-analysis of 18 articles dealing with screen time (TV, computer, phone, etc.) exposure by young children who are developing language and other skills. Electronic devices to which children are exposed include broadcast television/ movies and interactive social media/ video games, the former passive and the latter active. The AAP has documented deleterious effects and discouraged use of screen time with less than two-year olds and has published guidelines for parents; the WHO has likewise issued similar guidance to parents of children up to age five. The current review covers the past 20 years, includes 32,274 children from five continents.
The study is a welcome contribution to the literature on the effects of screen time on language development; the cited articles all show detrimental influence of screen time (especially of passive time) on both behavioral measures of language and other developmental milestones, and of cerebral development, as measured by neuroimaging. Table 1 is very clear and informative.
The discussion raises several important points that are explored in terms of the 18 articles, for example, the difference between passive and active screen time. The importance of parental intervention is underlined, as it is crucial for social as well as linguistic learning. Furthermore, working memory, attentional and executive functions are negatively affected, leading potentially to later difficulties in academic settings.
The overview seems to mainly focus on passive viewing, correctly noting that “future studies will increasingly need to focus on the digital media that children possess early” (l. 636-637). The development of interactive media (phones, tablets) has the potential of expanding interpersonal communication (e.g., using Zoom, Facetime, Whatsapp to connect with long-distance family members); furthermore, there are numerous new “toys” that encourage interaction using synthesized speech. Communicating with a real grandparent on a phone seems quite different than interacting with a toy that plays songs or pronounces words. These seem to be areas that will need to be explored.
This submission is a meta-analysis of articles dealing with infant development and exposure to "screen time". As such, it is not original research, simply a summary of 18 relevant articles (selected from an initial group of 278 articles). There is then no original premise, data collection, methodology, analysis or conclusions drawn from an original experiment. The article is valuable in providing an overview of a broad range of research on 0-5 year olds exposed to television, tablets and phones. It is not appropriate if the journal does not want to publish meta-analyses. Its conclusions show that the vast majority of articles from the past 20 years dealing with this topic clearly demonstrate that passive screen time is deleterious to linguistic, cognitive, small and large motor development. It would be helpful if the authors could provide more information on interactive screen time since the majority of the articles deal with passive screen time.
Reviewer 2 Report
Comments and Suggestions for Authors
Dear authors, Your manuscript addresses a compelling and highly significant issue. However, a comment should be provided.
The Introduction contained an excessive amount of non-scientific information, including several orders from AAP and WHO. It appears to be unrelated to scientific evaluation. However, the current examination of pertinent scientific material regarding the relationship between ST and language and cognitive development studies is insufficient.
The text lacks sufficient information regarding the inclusion and exclusion criteria used for selecting papers for review. It is unclear why 260 papers were eliminated. Why were the papers listed below not included?
The article explores various subjects, including the correlation between screen usage and cognitive development, linguistic development, and social development. But the search request was limited to: (screen time) AND (LANGUAGE 136 'ACQUISITION'). The corpus of literature used for discussing the relationship between ST and cognitive and social development was not sufficiently representative.
In the results section, there is a noticeable difference in the organization of text between point 3.2, which focuses on language, and point 3.3, which focuses on cognitive growth. The former is structured similarly to an abstract, whereas the latter appears to be an analytical review. It would be preferable to ensure that they are structured in a similar manner. Additionally, the data pertaining to language development should be more extensive and encompass more thorough information regarding distinct domains of language development, such as lexicon, grammar, and verbal activity. This section of text redundantly replicates the identical information found in the table. It would be more effective to avoid duplicating the data and instead offer it in a more evaluative and integrative manner.
The offered overview does not differentiate between the data received from direct children assessment and from parent questionnaires, in both the cross-sectional and longitudinal data. This disparity is significant.
The Discussion section addresses the data that was not supplied in the Result section, and the relevant goals were not presented in the Method section:
Line 486: 4.6. Brain and prolonged screen time: structural alterations in white matter and influence on language acquisition. This topic is not relevant to aims and obtained data presented in Result; it should be removed or the relevant data should be added to Method and Result;
Line 595: 4.9. Screen time and the development of social skills. This topic is not relevant to the aims and obtained data presented in Result; it should be removed or the relevant data should be added to Method and Result.
In summary, it is recommended that the manuscript undergo revision.

Reviewer 3 Report
Comments and Suggestions for Authors
The study focuses on analyzing scientific publications regarding the impact of gadgets on children's language development. The authors conducted a detailed and systematic review, summarizing the findings of relevant publications. The primary research question was: how does screen exposure to gadgets affect early childhood language development?
It's crucial that the authors chose young children for this study. Nowadays, children are engaging with gadgets earlier and earlier. However, many publications note a delayed speech development in contemporary children. Hence, this article's theme is highly relevant, addressing a specific gap in modern scientific knowledge.
Despite the presence of individual experimental studies, this article contributes by consolidating the existing findings to date.
The conclusions of the article entirely align with its initial positions and objectives. The sources used are appropriate and sufficient.
As a minor suggestion, I would advise the authors to include a bit more information about the neurological mechanisms of the studied phenomenon.
***
Best regards,
Your reviewer.
